# Exploring Psychological Factors for COVID-19 Vaccination Intention in Taiwan

**DOI:** 10.3390/vaccines9070764

**Published:** 2021-07-08

**Authors:** Shih-Yu Lo, Shu-Chu Sarrina Li, Tai-Yee Wu

**Affiliations:** 1Institute of Communication Studies, National Yang Ming Chiao Tung University, Hsinchu 30010, Taiwan; shuchu@mail.nctu.edu.tw (S.-C.S.L.); taiyeewu@nycu.edu.tw (T.-Y.W.); 2Institute of Communication Studies, National Chiao Tung University, Hsinchu 30010, Taiwan

**Keywords:** COVID-19, vaccination intention, conspiracy theories, mental models, powerlessness

## Abstract

To underpin the psychological factors for vaccination intention, we explored the variables related to positive and negative attitudes toward COVID-19 vaccination in Taiwan. The data were collected via an online survey platform with a sample size of 1100 in April 2021. We found that people’s interpretations of the origin of the virus were relevant. People who tended to believe that the virus was artificially created felt powerless and were more concerned about the possible side-effects of the vaccines, which was negatively associated with their vaccination intention. The source of vaccine recommendation was found to be relevant to vaccination intention. People’s vaccination intention was highest if the vaccines were recommended by health professionals, followed by friends and the government, and then mainstream media and social media. The analysis of the demographic variables showed that men tended to be more receptive to vaccines than women. Our findings should provide insights into developing communication strategies to effectively promote vaccination intentions.

## 1. Introduction

As the coronavirus disease (COVID-19) continues raging in much of the world, increasing the vaccination rate is considered an effective strategy to end the pandemic. However, the vaccination rollout is still low in many countries. For example, in Taiwan, only about 1% of 23 million people had received a shot by the end of May 2021 [1]. To accelerate the vaccination rollout, pinpointing possible factors that could affect vaccination intention is essential.

The intention to receive COVID-19 vaccines has been studied in multiple countries, such as Portugal [2], China [3], Indonesia [4], Ireland [5], Japan [6], UK [7,8], and the US [9], most of which approached this issue from a public health perspective and have thus focused more on the demographic features. In a study by Sherman et al. [7], who took a psychological perspective, the researchers integrated theories including the health belief model [10], the theory of planned behavior [11], and protection motivation theory [12]. They measured multiple constructs extracted from these theories based on 1,500 UK adults and found that the two major components were “General COVID-19 vaccination beliefs and attitudes” (*r*^2^ = 19.71%) and “COVID-19 vaccination adverse effects” (*r*^2^ = 8.18%). The former represents general attitudes toward the effectiveness of the vaccines, whereas the latter represents concern for the negative side effects induced by the vaccines. This study also measured the participants’ “general” vaccination beliefs and attitudes, but the explanatory power was only approximately 2%. The implication was that vaccination attitudes per se could not sufficiently account for the intention for COVID-19 vaccination.

### 1.1. Mental Models for COVID-19

We hypothesize that the mental models for the origin and way of spreading of COVID-19 should be an important factor for vaccination intention. The concept of the “mental model” originated in cognitive psychology and was used to theorize the process of deductive reasoning [13,14,15]. A mental model refers to the way people interpret how things work in the real world. Different mental models indicate different ways of representing events in the world, which prompt people to make different decisions or judgments. Take COVID-19 for example; people develop their own explanations for the origin and the way of spreading the virus. One type of mental model is a conspiracy belief, referring to people’s belief that an event is triggered by secret acts of powerful, malevolent forces [16,17,18,19,20]. The conspiracy theories surrounding COVID-19 included that the virus was created and spread by the Central Investigation Agency (CIA), the media, 5G technology, a biochemical laboratory in China, the US government, pharmaceutical companies, etc. [21].

Conspiracy beliefs have been shown to be negatively associated with COVID-19 vaccination intention, according to studies conducted in France [22], the UK [23], Italy [24], and other countries, including Ireland, Spain, the US, and Mexico [25]. The underlying psychological mechanism for this conspiracy-induced vaccination skepticism could be mediated by political powerlessness [21]. When people feel powerless, they feel they can do nothing to change the situation, and thus, they are less likely to take any action to prevent the spread of the disease. A similar phenomenon was also observed for people’s intentions to overcome climate change [26] and HIV/AIDS [27].

### 1.2. Aims and Hypothesis

In the present study, we aimed to test whether mental models for COVID-19 modulate people’s intentions to receive COVID-19 vaccines in Taiwan. As cultural orientation might be a factor in the tendency to endorse conspiracy theories [21], previous studies conducted in other nations might not apply to Taiwan. In fact, in a study conducted in Finland, belief in conspiracy itself was not significantly associated with vaccination intention [28].

In the study of Biddlestone et al. [21], the “conspiracies” included the ill-intended powers of the government, the CIA, 5G technology, or the biochemical lab in Wuhan, China. Interestingly, the Cronbach’s alpha was 0.88 among the 10 items that were designed to test different types of conspiracy beliefs. The high value of the reliability index implied that if one believed the origin of the virus to be from one source, for example the CIA, they also tended to believe it to be from a different source, for example, a lab in Wuhan. The common feature among these types of conspiracy beliefs was the artificialness about the origin or the way of spreading of the virus. Therefore, we designed two sets of questions that measured people’s mental models for the origin and the way of spreading of COVID-19. For each set, one question measured the degree of artificialness, and the other measured the degree of naturalness of participants’ thoughts about the virus. The exact questions were: *“I think the coronavirus was artificially produced, and spread deliberately,” “I think the coronavirus was artificially produced, and spread accidentally,” “I think the coronavirus emerged naturally, and spread deliberately,”* and *“I think the coronavirus emerged naturally, and spread accidentally.”*

We hypothesized that the different mental models for COVID-19 could lead to different degrees of powerlessness, which would modulate their attitudes and intention for COVID-19 vaccination. The items to measure the attitudes were adopted from the two major constructs associated with vaccination intention found in a previous study by Sherman et al. [7], which were “general COVID-19 vaccination beliefs and attitudes” and “COVID-19 vaccination adverse effects”. Some items in the former construct included other-dependent attitudes, such as *“If a coronavirus vaccination were recommended by the government, I would get vaccinated”* and *“If a coronavirus vaccination were recommended by a health care professional, I would get vaccinated.”* We moved these items to another category of “recommendation source,” because these items reflected people’s susceptibility to social influence on making a medical decision, instead of their attitudes toward vaccination. We further added possible recommendation sources, including friends, mainstream media, and social media. By comparing people’s intentions among different recommendation sources, we could have a clear idea of whom we should target to promote vaccination. The full model of our hypothesis is illustrated in Figure 1.

## 2. Materials and Methods

### 2.1. Participants

The research protocol was approved by the Research Ethics Committee for Human Subject Protection of National Yang Ming Chiao Tung University (#NCTU-REC-109-122W). The data were collected from an online survey with ETtoday—a contract marketing research company that excels in nationwide public opinion polls. The sample was deliberately chosen to match the distributions of different genders, cities/counties of residence, and age groups of the total internet users in Taiwan [29]. Responses were collected between April 1, 2021 and April 18, 2021 from a valid sample of 1100 respondents.

### 2.2. Materials

Six questions were designed to collect demographic information regarding the respondent’s gender, age, city/county of residence, level of education, monthly income, and occupation. Among a total of 80 other questions in the questionnaire, 24 (see Appendix A) were relevant to the present study. Four items were used to measure people’s mental models of virus. Three items were used to measure powerlessness, directly translated from items in the study by Biddlestone, et al. [21]. Seven and four items were used to measure “attitude toward the COVID-19 vaccination” and “COVID-19 vaccination adverse effects”, respectively, and were translated from the items used in the study by Sherman, et al. [7]. We selected items that had a loading value larger than 0.6. The six items used to measure vaccination intention included one item that measured people’s general intention to receive a vaccine. The other five items measured people’s tendency to receive a vaccine recommended or promoted by health professionals, friends, the government, social media, and mainstream media. All respondents rated each item on a 7-point Likert scale.

## 3. Results

### 3.1. Demographic Factors 

We first examined how demographic variables were associated with general vaccination intentions. With regards to gender (Figure 2), the data showed that men (*M* = 4.41, 95% CI 4.25–4.56) had a higher intention than women (*M* = 3.75, 95% CI 3.61–3.90) to receive a vaccine, *F* (1,1098) = 37.01, *p* < 0.001, *η*^2^ = 0.03. Other demographic variables, including age, *F* (8,1091) = 0.99, *p* = 0.44, *η*^2^ = 0.007, education levels, *F* (4,1095) = 0.73, *p* = 0.57, *η*^2^ = 0.003, income, *F* (10,1001) = 1.46, *p* = 0.15, *η*^2^ = 0.01, city/county of residence, *F* (21,1078) = 0.54, *p* = 0.95, *η*^2^ = 0.01, and occupation, *F* (15,1084) = 1.40, *p* = 0.14, *η*^2^ = 0.02, were not significantly associated with vaccination intention. The vaccination intentions of the different genders and age groups are illustrated in Figure 2.

### 3.2. Recommendation Source

The question *“If COVID-19 vaccines are available in Taiwan, I will get vaccinated”* was designed to test people’s general vaccination intention. Furthermore, five other questions were designed to test people’s intention to receive a vaccine if suggested or promoted by health professionals, friends, the government, social media, or mainstream media. We then compared participants’ intentions among the six items by a repeated-measures ANOVA with Greenhouse–Geisser correction, because the raw data violated the sphericity assumption. We found that the rating values differed significantly among them, *F* (4.64,5101.58) = 74.14, *p* < 0.001, *η*^2^*_partial_* = 0.06. People’s vaccination intention was highest when vaccination was recommended by health professionals (*M* = 4.46), followed by general intention (*M* = 4.09), friends (*M* = 4.02), the government (*M* = 4.00), the mainstream media (*M* = 3.89), and social media (*M* = 3.83). After incorporating the significance test results of the LSD post hoc analysis, vaccination intention could be most effectively promoted by health professionals, followed by friends and the government, and then both types of media. The general vaccination intention was equivalent to the vaccination intention if recommended by friends, and the comparison between friends and the government, and between the mainstream media and social media, did not yield a significant difference.

### 3.3. Attitudes toward Vaccination

The seven items in “attitude toward the COVID-19 vaccination” and four items in “COVID-19 vaccination adverse effects” were subjected to a factor analysis, with the principal axis factoring method of factor extraction followed by a varimax rotation (Table 1). Three factors were extracted based on the criterion of eigenvalue larger than one. Seven items in “General COVID-19 vaccination beliefs and attitudes” were loaded highly under one factor, and we renamed it the “positive attitudes for COVID-19 vaccination”; the three items in “COVID-19 vaccination adverse effects” were loaded highly under another factor, which we renamed the “negative attitudes for COVID-19 vaccination”. The third factor was only loaded with one item, *“A coronavirus vaccination could give me coronavirus,”* which we renamed “belief of vaccine-induced infection”.

According to our hypothesis, as illustrated in Figure 1, attitudes toward vaccination should be associated with vaccination intention. To test this, we then used the three factors of vaccination attitudes as independent variables, with general vaccination intention as the outcome variable, in a multiple regression analysis. Based on a bidirectional stepwise method, the factors of positive and negative attitudes toward COVID-19 vaccination were selected into the model, yielding an overall *r*^2^ of 0.67. The standardized regression coefficient was 0.77, *t* (1097) = 42.56, *p* < 0.001, and −0.12, *t* (1097) = −6.56, *p* < 0.001, respectively for the positive and negative attitudes (Table 2). Belief of vaccine-induced infection was not selected into the model due to its low explanatory power.

### 3.4. Mental Models for COVID-19 

The four items used to measure the mental models for COVID-19 were subjected to a factor analysis, with the principal axis method of factor extraction followed by a varimax rotation. Two factors were extracted based on the criterion of an eigenvalue larger than one. The first factor accounted for 28% of the variance, including *“I think the coronavirus was artificially produced, and spread deliberately”* (loading: 0.72) and *“I think the coronavirus was artificially produced, and spread accidentally”* (loading: 0.49), and the second factors accounted for 26% of the variance, including *”I think the coronavirus emerged naturally, and spread deliberately”* (loading: 0.75) and *“I think the coronavirus emerged naturally and spread accidentally”* (loading: 0.69). The results suggested that respondents had consistent opinions about the origin of the virus but not about the way of spreading of the virus. We then split the construct of the mental model into the two factors of “belief in artificial origin” and “belief in natural origin.”

According to our hypothesis, as illustrated in Figure 1, mental models for COVID-19 should be associated with attitudes toward COVID-19 vaccination. We then used beliefs in natural origin and artificial origin as two independent variables in a bidirectional stepwise regression analysis (Table 2) and found that both effects of belief in artificial origin, *β* = 0.21, *t* (1097) = 7.14, *p* < 0.001, and natural origin, *β* = 0.14, *t* (1097) = 4.64, *p* < 0.001, on positive attitudes toward COVID-19 vaccination were significant. For negative attitudes toward COVID-19 vaccination, only belief in artificial origin played a significant role, *β* = 0.07, *t* (1098) = 2.16, *p* = 0.03, while belief in natural origin was not included in the regression.

### 3.5. Mediation Analysis

Biddlestone, et al. [21] showed that belief in conspiracy theories decreases the intention for disease-preventing behavior via powerlessness. Thus, we used separate regression analyses to test whether powerlessness (*Cronbach’s α* = 0.82) was associated with positive and negative vaccination attitudes (Table 2), and we found its significant effect on negative attitudes, *β* = 0.18, t (1098) = 6.04, *p* < 0.001, but not positive attitudes, *β* = 0.05, *t* (1098) = 1.51, *p* = 13.

Therefore, the effect of belief in artificial origin on negative attitudes toward COVID-19 could potentially be mediated by powerlessness. We then used the SPSS PROCESS macro v 3.5 [30] to examine the mediation effect (model 4) with a bootstrapping method based on 5,000 times of resampling. Four conditions of mediation analysis were met (Table 3): First, belief of artificial origin was significantly associated with powerlessness, *t* (1098) = 5.33, *p* < 0.001; secondly, powerlessness was significantly associated with negative attitudes, *t* (1097) = 5.76, *p* < 0.001; thirdly, the total effect of belief of artificial origin on negative attitudes was significant, *t* (1098) = 2.16, *p* = 0.03; lastly, the 95% bias-corrected confidence intervals (CI) of the indirect effect from belief in artificial origin to negative attitudes via powerlessness was between 0.01 and 0.04, which did not include zero, supporting a mediating role of powerlessness.

## 4. Discussion

### 4.1. Demographic Factors of Vaccination Intention

We first examined the demographic variables associated with vaccination intention, and we found a significant effect of gender. In fact, previous research has shown that men are more inclined to adopt pharmaceutical interventions [31], including vaccination [32,33,34] than women. For COVID-19 vaccination intention, a multi-country study also reported that the rejection of vaccination was more than twice as common among women than among men [35]. With regard to the other demographic variables, their effects on vaccination intention were minimal. However, we were reserved concerning the absence of a significant age effect, as our data were collected online. For the senior age groups, the sample might not be representative because using the internet is still uncommon among them [29].

### 4.2. Recommendation Source

According to the comparisons among vaccination intentions promoted by different recommendation sources, people tended to follow the advice given by health professionals, followed by friends and the government, and then, the media. The tendency to endorse recommendations from health professionals is undoubtedly an adaptive behavior, but information from other sources, such as friends, the government’s propaganda, or the media, is much easier to obtain than information from health professionals. Among the three sources that are easy to approach, recommendations from both types of media were the least endorsed, and recommendations from friends were slightly prioritized over the government, with a statistically insignificant trend.

According to the theory of planned behavior [11], subjective norms, indicating a person’s beliefs about whether people of close relationships think they should engage in a certain behavior, is a variable strongly associated with behavioral intention. The importance of subjective norms can also be observed in other health-related behavior. For example, perceptions that peers were changing their behavior were consistently related to future HIV-preventive measures [36]. The fact that people give high credibility to health-related information from friend circles could potentially lead to a dangerous consequence: if misinformation about the virus spreads across society, it could be a barrier for the government or the media to debunk misinformation.

### 4.3. Mental Models

Belief in conspiracy theories has always been a very important factor of people’s intention to engage in disease-preventing behavior. In the present study, we incorporated belief in conspiracy theories into the “mental models” for COVID-19, as conspiracy beliefs can be viewed as one form of a mental model. We separated people’s mental model of the origin and the way of spreading of the virus, but the results of factor analysis suggested that the factor structure accorded with the origin instead of the way of spreading of the virus. How belief in natural origin and belief in artificial origin are associated with attitudes toward COVID-19 vaccination can be illustrated in Figure 3.

Both the beliefs in natural origin and artificial origin were positively associated with positive attitudes toward the vaccination. Possibly, people who rated highly in either of the two beliefs were people who felt more certain about the origin. It was probably also because of their tendency to feel certain that made them endorse the positive effect of vaccination. On the contrary, the uncertainty perceived by people who rated low in both types of beliefs was probably the cause of their low endorsement of vaccination. To test such speculation, we averaged the scores of belief in natural origin and belief in artificial origin as an index of certainty of the origin, and found it to be significantly associated with positive attitudes toward COVID-19 vaccination, *β* = 0.23, *t* (1098) = 7.64, *p* < 0.001.

Negative attitudes toward vaccination were only associated with the belief in artificial origin but not natural origin. This finding echoes previous studies that showed a negative effect of conspiracy belief on health-promoting behaviors [21,26,27,37], including COVID-19 vaccination [22,23,25]. One important difference between our study and previous studies was that we did not use any concrete “conspiracy” in our survey. Our findings suggest that it was the “artificialness” or “unnaturalness” quality of the virus that induced the feeling of powerlessness. The feeling of powerlessness might have raised participants’ defensive sentiments toward the authorities, intensifying their attention to the potential negative consequences of vaccination.

### 4.4. Research Limitations and Practical Implications 

The data of the present study were collected online, and therefore, the sample was biased toward internet users instead of the general public. According to a survey conducted in 2019 [29], the proportion of people who used the internet was 86.2% across the general population over the age of 12 in Taiwan. For people under the age of 59, the proportion was 96.7%. Therefore, we think that our findings are representative of the citizens who play an important role in policy making in Taiwan.

Another limitation is the timeliness of our findings. The data of the present study were collected in early April 2021, when Taiwan’s daily number of infected cases was as low as less than 10. However, in mid-May, the B.1.1.7 “UK” variant emerged in Taiwan, and the number of infected cases and death toll surged. Therefore, the current vaccination intention was probably much higher than the reported intention in this study. However, how high is enough? An important parameter for estimating the threshold of herd immunity is R0, which refers to the average number of infections caused by a single infected individual. Since the outbreak of COVID-19, various studies have estimated the R0 index to be in the range of 2 to 6 [38]. The threshold of herd immunity is estimated to be 50–83%, based on the rule that the vaccination rate should exceed the criterion of 1–1/R0 [39]. Taking neighboring countries such as Japan and Korea, which faced the same level of severity much earlier than Taiwan as references, their vaccination rates were 6.41% and 9.14% in late May [40], respectively, which are still far below the minimum requirements for herd immunity. Therefore, we believe that investigating the possible factors that promote vaccination intention is still important in Taiwan.

## 5. Conclusions

We followed the line of research [8] that integrated the health belief model [10], theory of planned behavior [11], and protection motivation theory [12] to underpin the psychological variables that could account for people’s COVID-19 vaccination intention in Taiwan. The vaccination intention varied depending on the recommendation sources, as well as positive and negative attitudes toward vaccination. For negative attitudes in particular, they were associated with the belief in the artificial origin of the virus, mediated by powerlessness. The findings of the present study could provide the government or health practitioners with an empirical base to design a strategy to promote vaccination.

## Figures and Tables

**Figure 1 vaccines-09-00764-f001:**
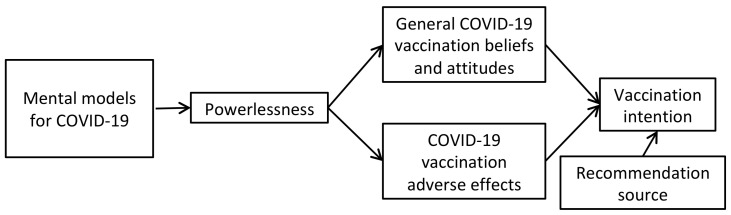
The hypothetical model for COVID-19 vaccination intention.

**Figure 2 vaccines-09-00764-f002:**
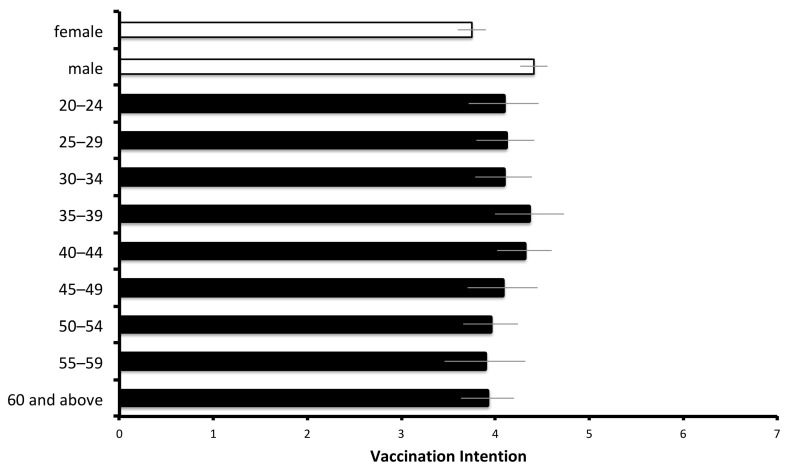
General vaccination intention of different genders, and different age groups. The error bar indicates the 95% confidence interval of the mean.

**Figure 3 vaccines-09-00764-f003:**
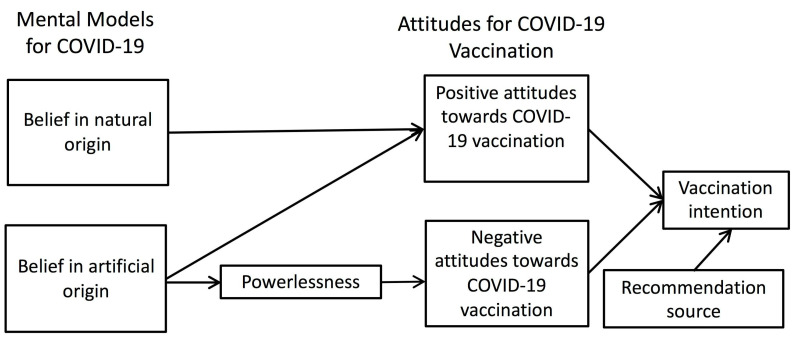
The full model supported by the data in the present study.

**Table 1 vaccines-09-00764-t001:** The loading value for each item measuring attitudes toward COVID-19 vaccination.

Construct	Items	Factor Loading
Positive attitudes toward COVID-19 vaccines (variance explained: 38%)	A coronavirus vaccination should be mandatory for everyone who is able to have it.	0.68
	If I get a coronavirus vaccination, I will be protected against coronavirus.	0.77
	If I don’t get a coronavirus vaccination and end up getting coronavirus, I would regret not getting the vaccination.	0.72
	Other people like me will get a coronavirus vaccination.	0.84
	My family would approve of my having a coronavirus vaccination.	0.80
	My friends would approve of my having a coronavirus vaccination.	0.77
	A coronavirus vaccine will allow us to get back to ‘normal’.	0.82
Negative attitudes toward COVID-19 vaccines	I would be worried about experiencing side-effects from a coronavirus vaccination.	0.74
(variance explained: 17%)	I might regret getting a coronavirus vaccination if I later experienced side-effects from the vaccination.	0.75
	A coronavirus vaccination will be too new for me to be confident about getting vaccinated.	0.75
Belief of vaccine-induced infection (variance explained: 4%)	A coronavirus vaccination could give me coronavirus.	0.39

**Table 2 vaccines-09-00764-t002:** Summary table of the regression analysis.

Independent Variable	Outcome Variable	*r* ^2^	*β*	Statistical Tests
Positive attitudes	Vaccination intention	0.67	0.77	*t* (1097) = 42.56, *p* < 0.001
Negative attitudes			−0.12	*t* (1097) = −6.56, *p* < 0.001
Belief in natural origin	Positive attitudes	0.05	0.14	*t* (1097) = 4.64, *p* < 0.001
Belief in artificial origin			0.21	*t* (1097) = 7.14, *p* < 0.001
Belief in natural origin	Negative attitudes	0.004	*n.s.*	*n.s.*
Belief in artificial origin			0.07	*t* (1098) = 2.16, *p* = 0.03)
Powerlessness	Positive attitudes	0.002	0.05	*n.s.*
Powerlessness	Negative attitudes	0.03	0.18	*t* (1098) = 6.04, *p* < 0.001

**Table 3 vaccines-09-00764-t003:** Summary of the mediation analysis.

Independent Variable	Outcome Variable	Notes	Statistical Test
Belief in artificial origin	Powerlessness		*t* (1098) = 5.33, *p* < 0.001
Powerlessness	Negative attitudes		*t* (1097) = 5.76, *p* < 0.001
Belief in artificial origin	Negative attitudes	Total effect	*t* (1098) = 2.16, *p* = 0.03
Belief in artificial origin	Negative attitudes	Indirect effect	95% CI 0.01–0.04

## Data Availability

The data presented in this study are available on request from the corresponding author.

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
