# Peer review of "Exploring Psychological Factors for COVID-19 Vaccination Intention in Taiwan"

_vaccines, 2021, doi:10.3390/vaccines9070764_

Round 1

Reviewer 1 Report

Dear Authors 

Your study essentially replicates other similar findings. As such I find it useful.

Please note that your study shows correlation, not causation so you cannot conclude anything for sure about the effects of better communication etc. Moreover, I would strongly suggest that you look into sentences like this:

"We are yet to know the actual origin of the virus, but emphasizing the possibility of artificial origin might need to be avoided, at least not at the moment when our primary goal is to promote vaccination intention" 

You seem to suggest a kind of undemocratic process in which government and experts may decide what information we may acces. This in my mind is far beyond the scope of a paper presenting the results of a survey

two main concerns 

  • correlation is treated a proof of causation
  • the recommandations are not so democratic 

Author Response

We thank the Reviewer for their time and effort in providing advice and insightful comments on our previous manuscript. Please see the point-by-point reply below.  

To Reviewer #1:

Dear Authors 

Your study essentially replicates other similar findings. As such I find it useful.

We thank the Reviewer’s compliments on our manuscript.

Please note that your study shows correlation, not causation so you cannot conclude anything for sure about the effects of better communication etc. Moreover, I would strongly suggest that you look into sentences like this:

"We are yet to know the actual origin of the virus, but emphasizing the possibility of artificial origin might need to be avoided, at least not at the moment when our primary goal is to promote vaccination intention" 

We thank the Reviewer for pointing out the confusion of correlation and causation in our previous manuscript. We have removed this sentence mentioned by the Reviewer.

You seem to suggest a kind of undemocratic process in which government and experts may decide what information we may access. This in my mind is far beyond the scope of a paper presenting the results of a survey

We have removed this “suggestion” in the revised manuscript.

two main concerns 

  • correlation is treated a proof of causation
  • the recommandations are not so democratic 

We have removed the relevant parts and made sure that the two types of mistakes have been cleared in the revised manuscript. The Reviewer can also find that in our revised manuscript, we have removed the words that implied a causal relationship such as influence, increase, or reduce.

Reviewer 2 Report

General

This study tried to predict the psychological factors for COVID-19 vaccination intention in Taiwan, incorporating the result of past studies as appropriate. In the study, it was demonstrated that vaccination intention could be predicted by the recommendation sources, as well as by positive and negative attitudes toward vaccination.

The models and statistical methods adopted in this study are well-balanced, from standard to challenging, and are evaluated from a psychological point of view. While this survey has a research limitation related to questionnaire target (senior age group), its design makes it useful for future surveys in other countries.

The authors need to make sure that they follow the journal’s guidelines for the notation of statistical symbols.

Line 231

What is the reason for choosing the bootstrapping method out of major resampling methods? Have you tried the jackknife method?

Author Response

We thank the Reviewer for their time and effort in providing advice and insightful comments on our previous manuscript. Please see the point-by-point reply below.  

To Reviewer #2:

This study tried to predict the psychological factors for COVID-19 vaccination intention in Taiwan, incorporating the result of past studies as appropriate. In the study, it was demonstrated that vaccination intention could be predicted by the recommendation sources, as well as by positive and negative attitudes toward vaccination.

The models and statistical methods adopted in this study are well-balanced, from standard to challenging, and are evaluated from a psychological point of view. While this survey has a research limitation related to questionnaire target (senior age group), its design makes it useful for future surveys in other countries.

The authors need to make sure that they follow the journal’s guidelines for the notation of statistical symbols.

We thank the Reviewer for reminding us about the journal’s requirements for presenting statistical results. We checked the “Instructions for Authors” section on the Vaccines website but could not find relevant information. Therefore, we consulted other articles in the same Special Issue, and modified our statistical reports accordingly.

Line 231

What is the reason for choosing the bootstrapping method out of major resampling methods? Have you tried the jackknife method?

We thank the Reviewer’s suggestion. We conducted the mediation analysis with the SPSS PROCESS macro v 3.5 (Hayes, 2017), as described on p. 7. The bootstrapping method is the default method in this package (Hayes, A.F., Introduction to mediation, moderation, and conditional process analysis: A regression-based approach. 2017: Guilford publications). We believe that there must be other ways to run the mediation analysis, but we just chose one that is well-cited in social sciences; we believe this method should be easier for our target readers to understand.